# A Novel Treatment: Effects of Nano-Sized and Micro-Sized Al_2_O_3_ on Steel Surface for the Shear Strength of Epoxy–Steel Single-Lap Joints

**DOI:** 10.3390/polym14173438

**Published:** 2022-08-23

**Authors:** Wanru Wang, Zhen Wang, Rui Guo, Guijun Xian

**Affiliations:** 1Key Lab of Structures Dynamic Behavior and Control of the Ministry of Education, Harbin Institute of Technology, Harbin 150090, China; 2Key Lab of Smart Prevention and Mitigation of Civil Engineering Disasters of the Ministry of Industry and Information Technology, Harbin Institute of Technology, Harbin 150090, China; 3School of Civil Engineering, Harbin Institute of Technology, Harbin 150090, China

**Keywords:** adhesion, interface mechanical properties, nanostructures, shear strength

## Abstract

Traditional steel surface treatment (e.g., sand blasting, or silane treatment) was regarded as an effective method to improve the bonding strength of steel–epoxy single-lap joints. In the present study, a new steel surface treatment method was developed. With this method, the steel surfaces were treated with suspensions of nano-sized and micro-sized Al_2_O_3_ particles in ethanol/water mixture using the dip-coating method. Both Al_2_O_3_ particle sizes were previously treated or not treated with silane. Single-lap shear tests of the steel–epoxy bonds were conducted to compare the effects of the treating methods. According to the testing results, the highest increase in the bonding strength (by 51.8%) was found for the steel coated with the suspension of silane treated nano-Al_2_O_3_ particles. Scanning electron microscopy (SEM) analysis and energy dispersive spectrometer (EDS) analysis indicates that the nano-Al_2_O_3_ particles were clearly attached to the treated steel surfaces. Moreover, the steel surface with the silane-treated nano-Al_2_O_3_ particles was found to clearly enhance the contact angle between the steel and epoxy resin. The fracture morphology analysis of the single-lap shear testing specimen shows that the bonding between the steel and adhesive changed from steel–epoxy interfacial failure to cohesive failure when the steel surfaces were treated with the nano-Al_2_O_3_ particles suspension. The developed steel surface treatment method with the suspension of nano-particles proves to be effective and reliable in enhancing the bonding strength of the steel-to-epoxy adhesives.

## 1. Introduction

Adhesive-bonded joints provide many advantages compared to traditional joints such as welding, bolting, and riveting. However, these joints suffer from a poor performance of the steel–epoxy interface [1,2,3,4,5,6]. One of the most potential treatments is adding nano-materials to the weak layer. Recent studies showed that different kinds of nano-particles can be coated onto the steel surface, such as fullerene, carbon nanotubes (CNTs), and graphene [7,8,9,10]. In this study, nano/micro aluminum trioxide (Al_2_O_3_) particles were both used in the experiments. Different contents of nano/micro particles were compared. For the reinforcement effect to be guaranteed, the main challenge regarding the incorporation of nanomaterials is the ability to achieve their uniform dispersion by disaggregation of the micrometric agglomerates formed as a consequence of attractive van de Waals interactions among the particles [11,12,13,14,15].

Some researchers have revealed that the enhancement efficiency of the covalent bond is more than five times than the Van der Waals force interface. In fact, the force between nano-materials and substrate was the Van der Waals force, resulting in the cracking of composites’ interface and lower properties [3]. Silane was regarded as a kind of effective chemical bond in the solution, and it can provide a strong covalent bond in the interfacial area. Although silane has been widely used in the engineering field, the degree of coverage and functionalization of surfaces upon silanization depends on the chemical functionality of the surface and silane. An optimum concentration of silane in the solution exists and yields an adequate molecular coverage on the surface of the functionalized filler. The optimum concentration is known to be in the range of 0.03~5 wt.%.

In the application area, the main problems are that silane-modified nano-particles can easily agglomerate in the solution. In particular, different sizes of particles will seriously determine the effect of the application. For example, in the field of interface bonding between steel and adhesives, nanoparticles act as the links between the steel/resin. The well-dispersed particles can be designed as the reinforced particles in the interfacial area. The size of the nanoparticles is the key factor affecting the dispersion and mechanical properties [16,17,18]. Al_2_O_3_ particles were regarded as the particle with the most potential in the nano-material application area. The sol–gel method was regarded as an efficient method to modify the particles in the solution [19,20,21]. Physical and chemical methods are the common methods used for nanoparticles. For smaller particles, chemical dispersants are more effective. Polyethyleneimine (PEI) and polystryrene sulphonic acid (PSS) are two strong acids that provide an effective function for the dispersion of Al_2_O_3_ particles [22].

The interface of the composite is one of the key area influencing the performance of the joints. İsmail Saraç et al. investigated the effects of Al_2_O_3_, TiO_2_, and nano-SiO_2_ of nano-particles in resin to enhance the properties of the single-lap joint. The results showed that the shear strength was improved up to 97% in 4 wt.% nano-Al_2_O_3_ reinforced specimens [23]. E. Marin et al. analyzed the intrinsic corrosion resistance of a TiO_2_/Al_2_O_3_ atomic layer on a grinded steel surface. The polarization curves showed that atomic layer coatings improved the corrosion resistance of stainless steel [14]. Iclal Avinc Akpinar et al. obtained the failure mode of adhesively bonded joints by using Graphene-COOH, Carbon Nanotube-COOH, and Fullerene C60 as the nanostructure in resin. 0.25%, 0.5%, 1%, 2%, and 3% nano-particles were examined in the experiments, and the results showed that 1% was the best nanostructure reinforcement ratio [24].

In this study, a suspension of silanized nano/micro-sized Al_2_O_3_ particles in a water/ethanol mixture was developed as the surface treatment for steel surfaces to improve the bonding performance between steel and epoxy adhesive. SEM, EDS, and contact angle tests were conducted to analyze the morphology and properties of the steel surface after treatment. The interfacial bonding properties were determined by the single-lap joint test method. The study aims to provide an effective nano-particle-treated method on steel surfaces to enhance the bonding performances of steel–epoxy joints in the civil engineering field.

## 2. Experimental

### 2.1. Raw Materials

Q235 steel plate was used in the experiment. The size of the steel plate is 102 mm × 26 mm × 2 mm. Room temperature curable epoxy resin (bisphenol-A, (DGEBA)) and curing agent with a weight ratio of 100:31 was provided by Shandong Dagong Composite Materials Co. Ltd., Linyi, China. Nano α-Al_2_O_3_ (the diameter is about 30 nm) and micron α-Al_2_O_3_ (the diameter is about 1 μm) was bought from Maoguo nano-technology company, Shanghai, China. The KH550 silane coupling agent (γ-amino-propyltriethoxysilane) was used (molecular formula is H2NCH2CH2CH2Si(OC2H5)3).

### 2.2. Steel Surface Treatment

In the present study, steel was treated with grinding treatment, sand blasting, silane, nano-Al_2_O_3_ coating, and micron-Al_2_O_3_ coating. Each treatment was done as follows (Table 1).

Grinding treatment: One end of steel plate (26 mm × 12.7 mm) was cross-grinded by sandpaper (120 mesh). The samples were considered as control sample and designated as “C”.

Sand blasting treatment: After grinding treatment, steel plates were sandblasted using a sand blasting machine. The nozzle of the spray gun is 10 cm from the steel plates in 30 s; the particle of sand is about 100 μm, which was designated as “S”.

Silane treatment: 2 wt.% silane agent was mixed into ethanol–water solution (weight ratio of 9:1) and magnetically stirred for 20 min at room temperature, and the solution was designated as “silane solution”. “S” sample was immersed by silane solution, and then the steel was dried in an oven at 110 °C for 1 h; the steel was designated as “Si”.

Al_2_O_3_ treatment: For the steel coated with Al_2_O_3_ suspensions, the first step was to prepare appropriate Al_2_O_3_ suspensions. Nano and micron Al_2_O_3_ of 1 wt.%, 2 wt.%, 3 wt.%, and 4 wt.% were added into the 9:1 mixture of ethanol and water, respectively. Nano and micron Al_2_O_3_ of 1 wt.%, 2 wt.%, 3 wt.%, and 4 wt.% were slowly added into the prepared silane coupling solution mentioned above at room temperature and magnetically stirred for 10 min, respectively. Next, the Al_2_O_3_ suspension was sonicated for 30 min at room temperature. The Al_2_O_3_ suspensions were then dip-coated onto the steel surface.

### 2.3. Preparation of Single Lap Joints

The prepared steel plates end (26 mm × 12.7 mm) were coated with epoxy resin. A prepared steel plate and a resin covered steel plate were overlapped. After that, the overlapping region was pressed by a weight of 400 g, as shown in Figure 1. A standard curing cycle involved being at room temperature for 1 day, followed by a post curing process at 60 °C for another day. The average thickness of the cured resin between steels was consistent and within the range of 0.15~0.25 mm.

### 2.4. Scanning Electron Microscopy (SEM) Analysis

The morphology of the steel surface was characterized by SEM, and the test voltage was 20 kV. Then, the steel was coated with a layer of gold on the surface by the precision etching and plating instrument (Gatan682, American Gatan Company, Pleasanton, CA, USA).

### 2.5. FTIR Analysis

The covalent bond of solution was tested by FTIR analysis (Spectrum 100, Perkin Elmer Instruments, Waltham, MA, USA) from wavenumbers of 4000 cm^−1^ to 400 cm^−1^.

### 2.6. Contact Angle Tests

The contact angle machine was provided by Daheng (Group) Co., Ltd., Beijing Image Vision Technology Branch, Beijing, China. The size of sample was 40 × 26 × t mm^3^. The samples were tested by the contact angle machine. For each group, five points were tested in order to obtain a reliable and repeatable result.

## 3. Results and Discussion

### 3.1. Al_2_O_3_ Suspension of Tests

Nano Al_2_O_3_ suspensions were well dispersed in alcohol in 2mins, as shown in Figure 2a. After 6 h, the nano solution remained in a well-dispersed state (Figure 2b). This kind of phenomenon can be attributed to the size of the particles and the dispersion processes, which mean that the nano-Al_2_O_3_ can easily disperse in the ethanol solution.

The size of particles plays an important part in the solution. Because the colloid is in constant motion, the molecules around it gain kinetic energy and can counteract gravity without settling. Because the specific surface area of the nano-particle is large, the particle in the solution is more susceptible to the electrostatic action of the surrounding liquid on the particle, thus, the dispersion occurs [25].

The dispersion process of nano-particles can be divided into four steps. The first step is incorporation, then infiltration, disintegration of particle groups, and agglomeration of dispersed particles. Nano-particles tend to diffuse in the ethanol solution, just like the solvent in a solution with a disordered state. The smaller the size of the nano-particles, the stronger the motion pattern will be. When the nano-particles are dispersed in the solution for a period of time, the particles in the solution easily reach equilibrium, and after several hours, the effect of agglomeration and dispersion will reach a balance. According to the DLVO theory, there are mainly two kinds of forces between colloid; one is inter-molecular force, which is mainly manifested as mutual attraction between colloid, and the other is electrostatic force, which is mainly manifested as mutual repulsion between colloid [26].

In the solution, these two forces work together, and when the attractive force is greater than the repulsive force, the solution will aggregate and deposit. When the repulsive force is greater than the attractive force, the colloid particles in the solution will maintain a higher energy barrier under the action of the repulsive force. When the Van der Waals’s force and inter-molecular force keep in equilibrium, the energy barrier in the solution is zero. Then, the solution system is extremely unstable. Polymerization occurs when nanoparticles collide [27].

The sol–gel system is a kind of metastable system. The higher the energy barrier is, the longer the stable time is. When the height of the energy barrier is zero, the system is extremely unstable and the particles meeting will aggregate and deposit [28].

In the process of ultrasonic, the nano-particles absorb energy. Then, the nano-particles are influenced by Brownian motion, but the system is very unstable. As time goes by, the number of nanoparticles influenced by Brownian motion is reduced and slows the movement of the particles. In order to reduce the high surface energy, particles will reunite through molecular inter-atomic forces and deposit together [9].

The diameter of the nano-particles was about 100 nano-meters; the mixture is called the colloid, with a relatively stable state. Brownian motion is an important reason for the stable dispersion of colloids. In the solution, the colloidal particles do not stay in a fixed position, so the colloidal particles do not deposit due to gravity. In colloid, colloidal particles agglomerate by colliding with each other. The particle dispersion system is a multi-phase dispersion system, and the surface free energy of the dispersion system increases when the size of particles decreases.

After a certain period of time, the colloidal particles will collide with each other and reunite. The colloidal particles will reunite from small to large, and this reunion will reach a state of equilibrium due to Brownian motion. Gravity cannot be ignored for particles with a large mass or of a particle size, such as micron particles on the micron scale. After a period of time, in the ethanol solution, the effect of gravity is greater than the dispersion of particles in the solution, so deposition occurs. These two kinds of nano-particles are both used for the application of steel treatment.

FTIR was conducted to test the covalent bond between Al_2_O_3_ and silane in solution. According to the FTIR analysis of the two kinds of solutions (Figure 3), large amounts of hydroxyl groups can be detected in the solution at about 3300 cm^−1^. This is due to the abundance of hydrolyzed silane molecules in the solution. Nano-Al_2_O_3_ formed the covalent bonding in the mixed solution. The covalent bond is located near the peak position of 878 cm^−1^. At about 1040 cm^−1^, the Al_2_O_3_ formed the covalent bonding with silane (Table 2).

### 3.2. Steel Surface Treatment

#### 3.2.1. Surface Topography of Steel by Different Treatments

After the procedure of sand blasting, the surface roughness of steel was enhanced. The morphology of the steel surface is shown in Figure 4a. However, different sizes of particles on the steel surface showed a variety of properties. As shown in Figure 4b–e, micro Al_2_O_3_ particles were coated on the steel surface, but due to its larger size, the particles cannot form stable nano/micro structures on the steel surface [29]. Thin silane molecules film could form strong covalent bonding between steel/particles. Nano-Al_2_O_3_ particles grew on the steel surface, which could form a good film structure [1].

As shown in Figure 5a, there was an obvious gully on the grinding steel surface. After sand blasting (Figure 5b), the steel surface roughness was highly improved and the regular gully disappeared. In Figure 6a–f, 1–3% silanized Al_2_O_3_ was shown in SEM photos. Compared with 1 wt.% Al_2_O_3_, in 3 wt.% nano-Al_2_O_3_ was an obvious reunion. Differently sized particles form different morphologies on the steel surface.

The uniform dispersion of nanoparticles in the resin was clearly observed. In Figure 6a,b, it is obvious that the amount of nano-Al_2_O_3_ was lower than that in Figure 6c,d.

The EDS photos (Figure 7) show that silanized Al_2_O_3_ particles were observed in elemental analysis. In addition to the obvious aluminum element, silicon can also be clearly observed in EDS analysis. It shows that the silanized nanoparticles were adsorbed on the surface of steel. It is further proved that the treatment of Al_2_O_3_ particles on the surface of steel is effective.

#### 3.2.2. Contact Angle Tests

On steel surface, contact angle is related to the roughness of the steel surface and the chemical composition. The greater the roughness is, the better the wettability of the water molecules at the interface is. Chemical treatment also affects the wettability of the liquid on the steel surface to some extent. This is because when different chemical components are grafted on the surface of steel, different functional groups and polarity of water molecules will produce different forces, which will lead to the formation of water molecules of different angles at the interface of micro droplets (in Figure 8).

For grinded steel, the contact angle between water and steel was only 67.8° ± 1.78, whereas the contact angle of diiodomethane and steel was 28.2° ± 3.04. The nano-Al_2_O_3_ particle-treated steel surface was 120.9° ± 4.38, whereas the contact angle of diiodomethane and steel was 48.1° ± 1.98. The results of the contact angle tests show that the hydrophobic properties were improved after the nano-grafting procedure. The micro-structure was formed on the steel surface by the sand blasting procedure and the gully of steel was only several micrometers. Moreover, after the nano-grafting procedure, the steel surface was covered by nano-particles. The hydrophobic properties are due to the micro/nano-structure of steel.

In the SEM photos (Figure 9a,b), it can be observed that the bottom surface of the triangular structure formed by sandblasting and the length of the peak is 12 μm. During sandblasting, the arrangement of the surface of the matrix is regular; the smallest circulating elements can be simplified into the structure shown in Figure 10b. Light green represents the lower surface of the droplets of distilled water.



cosθ_r_ = f_1_cosθ_1_ + f_2_cosθ_2_
(1)



According to Equation (1), θ_r_ is the intrinsic contact angle between a solid and a liquid, and θ_1_ is the solid-liquid contact angle. The grinded samples followed Young’s equation and θ_r_ = 67.80 ± 1.7°, θ_1_ = 120.9 ± 4.38° and f_1_ + f_2_ = 1. According to Equation (2), f_1_ can be obtained as 35.36%. The Cassie–Baxter model and the triangular size formed by sandblasting can infer that the droplet depth of the distilled water covering the steel substrate is 1.46 μm.
(2)$$f_{1}=\frac{{\cos\theta }_{r}+1}{{\cos\theta }_{1}+1}$$

In Figure 11, Young’s equation is applicable to the solid–gas, solid–liquid, liquid–gas three-phase interface, namely, the ideal solid surface with a smooth surface and uniform composition, and the tension between the contact angles accords with the equation. However, the equation only applies to the ideal smooth surface; to investigate the actual production of the roughness of the surface, Wenzel et al. carried out thorough research and point out that when the roughness is more, the solid and liquid in the actual contact area is greater than the ideal smooth surface, formed in the substrate surface hydrophobic or hydrophilic surface; thus, the Wenzel equation was proposed.

When the composition of a solid surface becomes more complex, the Wenzel equation is no longer applicable. On this basis, Cassie et al. proposed the air rough cushion model. At the same time, in the actual micro–nano structure surface, sometimes the surface is metastable because it is still in an unstable state.

In other words, the liquid does not completely infiltrate into the microstructure on the solid surface, and there is a certain gap between the solid–liquid materials, which makes the three phases form a new composite model. The Cassie–Baxter state refers to the model in which the liquid droplets do not completely immerse into the microstructure, and there is a small amount of air residual, which is also the micro/nano structure model applicable to this paper.

As an important indicator to measure the hydrophobic performance of solid surfaces, the contact angle of the steel surface after grinding was 67.8° ±1.78 in this paper. As the substrate surface was relatively flat, the contact area between the steel surface and the substrate is close to the actual contact area when water droplets drop on the substrate surface, and Young’s model was adopted.

After the sandblasting treatment, the contact angle between the sample and the substrate surface was reduced to 33.5° ± 4.76. After spraying, silanized nano-coating is coated on the surface of the sample, and an air transition layer can be added between the silanized nano-coating and the droplet, so that the droplet will not make full contact with the steel matrix, thus improving the hydrophobic performance of the steel surface to a large extent. For example, the steel matrix treated by 2 wt.% Al_2_O_3_ nano-particle suspension.

The contact angle with water droplets was increased to 102.7° ± 1.89. If the cause of a solid surface roughness microstructure size in the submicrometer or nanometer order of magnitude, and the solid material and liquid contact angle value is above 90°, at this point, the microstructure in the air will not squeeze out a liquid but will be surrounded by a liquid and trapped within the microstructure, and will fill the microstructure airspace. This creates a composite surface where air and solid surfaces intersect. The presence of air will further improve the value of the apparent contact angle. It can be said that any solid surface with an apparent contact angle greater than 120° with water has such a microstructure, which is a composite surface formed by air and solid materials, such as the contact angle of 120.9° ± 4.38 after 3 wt.% Al_2_O_3_ particle treatment.

Silane coupling agent molecules with active groups on the surface of the steel adsorbed the substrate, so as to form a relatively dense structure on the side that is close to the substrate of the silane coupling agent molecular film, and in the process of adsorption, the formation of the structure became looser, and the formation of the silane coupling agent film leads to the relatively uniform dispersion of nanoparticles.

Due to the polymerization characteristics of the silane coupling agent itself, covalent bonds are easily formed between the active hydroxyl groups, whereas the unreacted amino group is exposed on the outside. Because the polarity of the amino group is weaker than that of the hydroxyl group, the contact angle formed between the amino group and the water drop is significantly higher than that of the steel matrix with only the active hydroxyl group in Figure 12.

Using the XRD analysis, it can be concluded that, compared with the sandblasting treatment in Figure 13, the presence of α-Al_2_O_3_ can be obviously detected on the substrate surface of sandblasting treatment and surface grafted Al_2_O_3_ film at about 27°. All of the three samples shows peaks of Fe on steel surface at about 44°, 65° and 83°.

In Figure 14, after soaking in 60 °C distilled water, floating rust occurs on the substrate surface of the sample after grinding, whereas only slight modifications occurs on the substrate treated with silanized nano-Al_2_O_3_ film. This corrosion extends from the edge of the sample to the center, indicating the good integrity of the silane film.

### 3.3. Interfacial Tests between Steel and Resin

#### Single-Lap Joint Tests

The single-lap test is a commonly used mechanical method to test the interfacial properties of steel/resin. The interface bonding properties of steel surface and resin treated by different methods can be tested through single-lap joint tests.

In Figure 14, the overall distribution of strain of the sample is symmetric along the width of the sample and anti-symmetric along the axial direction of the sample. However, due to the uneven force of the sample in the loading process, the strain value is slightly asymmetrical.

During the measurement of interfacial shear force in the single-lap sample, it is obvious that the strain value on the surface of the sample is at its maximum at about 5 cm from the edge of the lap wire. The maximum strain value can reach 0.0615 for the control group sample and 0.00515 for the sample treated with aluminum oxide particles.

The single-lap specimens showed that the single shear strength of the specimens gradually increases by 16.5% after the treatment of the steel surface by nano-Al_2_O_3_, and the interfacial bonding properties of the specimens can be further improved by 52.8% by silanized nano-particles treatment (Figure 15a). The detail are shown in Table 3.

The micro-particles were also tested by universal tensile testing machine. The single lap specimens shown that the single shear strength of the specimens gradually increases by 27.2% after the treatment of the steel surface by micro-particles, and the interfacial bonding properties of the specimens can be further improved by 16.5% by silanized micro-particles treatment [11] (Figure 15b). The detail are shown in Table 4.

DIC morphology of the single-lap sample is shown in Figure 16. The green area is the test cloud area, and the line shown in the figure is the test line. There were 200 spaced points on the line to be tested (Figure 16). White paint and black paint speckle were sprayed on the surface of the sample by VIC-3D technology. Then, the deformation of speckle on the surface of the sample under different load levels was observed by camera and LED technology, so as to analyze and compare the influence of different treatment methods on the mechanical properties of the single-lap samples.

Figure 17 shows the 2D picture of the control sample and 2 wt.% Al_2_O_3_ sample. In Figure 18 (based on Figure 16), when the load level is 90% Pu, the strain value of the control sample is 0.004; when the load level is 100% Pu, the strain value is 0.014, which means that when the load level of the sample increases by 10%, the strain value increases by 3.5 times. As shown in Figure 19 (based on Figure 16), the maximum load value of the control group sample is 6 KN, and the maximum load value of the nano-treated sample is 8 KN.

Adding a certain amount of nano-particles into the resin can effectively improve the bonding property of the steel–resin interface. But in the resin system and steel used in this paper, the increase of nano-particle content does not significantly affect the interface properties (from 1–4%); it has little effect on the bonding property of the steel–resin interface [12].

In Figure 20a, the epoxy could directly bond with steel using nano-Al_2_O_3_. In Figure 20b, the sketch of epoxy resin and steel bonding performance shows that silane could form a thin film between the steel and resin [12,13].

In the optical photos shown in Figure 21, the fracture was made relatively smooth by grinding. After tensile treatment, the fracture surface of the steel treated with Al_2_O_3_ particles shows a different morphology. The resin morphology of the single-lap fracture can be observed by optical microscope, and it further confirmed that the failure mode of the interface changed after different treatment methods. The fracture morphology of the sample surface can be clearly observed by optical microscope. After nano-Al_2_O_3_ treatment, the weak layer of the single-lap joint sample was transferred from the steel resin interface to the inside of the adhesive layer [9].

After the grind treatment, the fracture of the single-lap sample was relatively smooth, and the failure was mainly caused by the failure of the interface between the resin and steel. Among them, 2 wt.% nano-Al_2_O_3_ particles showed the most obvious effect on improving the interfacial bonding property, and obvious corrugated cracks can be observed at the fracture (in Figure 22). This phenomenon indicated that the fracture at the interface of the sample after nano-treatment was significantly improved. Adding nano-particles into resin can improve the bonding property of the interface to a certain extent, but the improvement effect is not as obvious as that of steel surface [14].

## 4. Conclusions

In the present paper, suspensions of nano/micro-Al_2_O_3_ particles in ethanol/water were used to treat the steel surface using the dip-coating method. By FTIR and SEM analysis, the silanized nano-Al_2_O_3_ can form a thin film by covalent bonding. The shear strength of nano-Al_2_O_3_-treated samples was improved by 51.8%. The shear strength of steel/adhesive joints is influenced by steel surface conditions related to particle dispersion state, surface roughness, and chemical bonding.

A new steel surface treatment method was developed. The steel was treated with a suspension of nano- or micro-Al_2_O_3_ particles in a mixture of ethanol/water using the dip-coating process. The particles were previously treated, or not treated, with silane. Compared to the method in which the adhesive is blended with nano-particles, the present dip-coating method showed better results in the enhancement of the interfacial shear strength between the steel—epoxy adhesive.Steel treated with the suspension including silane-treated nano-Al_2_O_3_ increases the bond strength between the steel—epoxy by 51.8%, much higher than those treated with the micro-Al_2_O_3_. For the later case, the increase is about 28.1%. Nano-Al_2_O_3_ is more likely to form multi-scale microstructures on the treated steel surface.Particles in suspension treated with a silane coupling agent was previously necessary to modify and functionalize the nano-particles and finally results in enhanced surface treatment effect. FTIR analysis shows that a silane coupling agent can form covalent bonds with Al_2_O_3_ particles. SEM analysis showed the silane coupling agent forming thin films with Al_2_O_3_ particles on the steel surface.

## Figures and Tables

**Figure 1 polymers-14-03438-f001:**
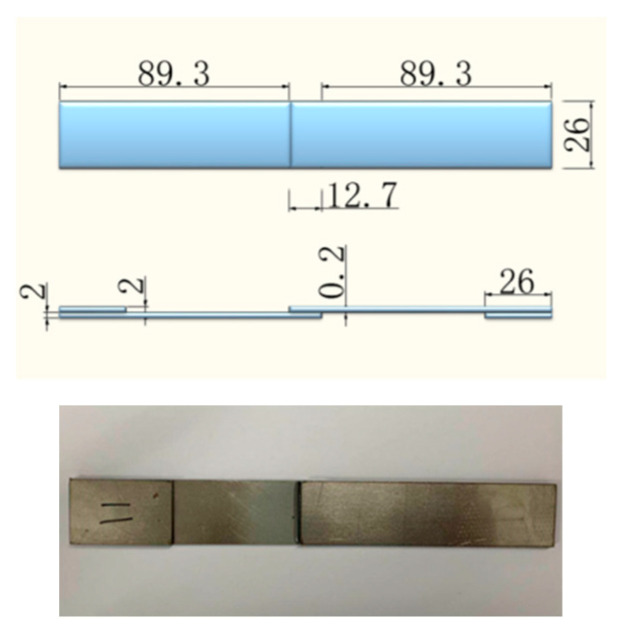
Size of single-lap samples.

**Figure 2 polymers-14-03438-f002:**
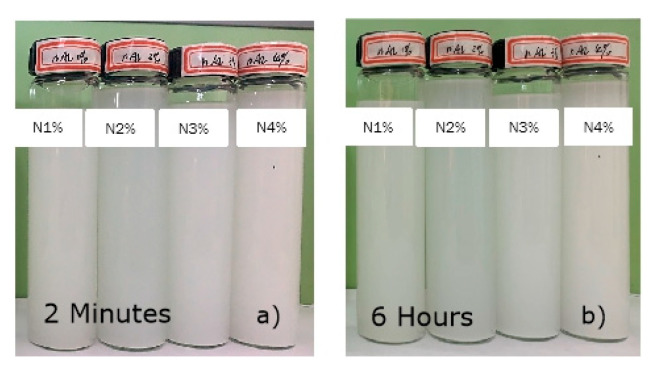
Photos of micro/nano-particles suspension solution in (**a**) 2 min. (**b**) 6 h.

**Figure 3 polymers-14-03438-f003:**
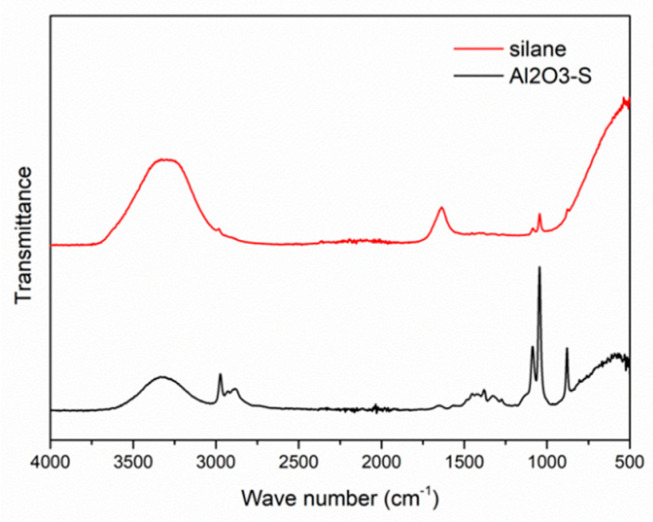
FTIR spectrum of silane solution and silanized Al_2_O_3_ suspension.

**Figure 4 polymers-14-03438-f004:**
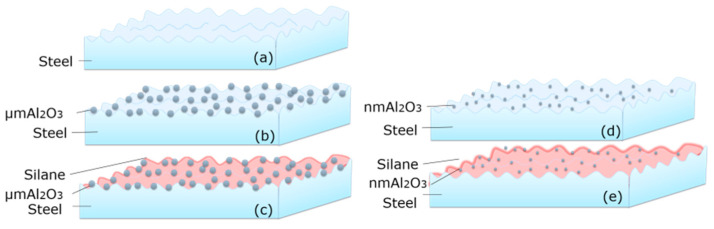
Schematic diagrams of steel surface morphology after (**a**) sand blasting. (**b**) Micro Al_2_O_3_; (**c**) nano Al_2_O_3_; (**d**) micro-silanized Al_2_O_3_; and (**e**) nano-silanized Al_2_O_3_.

**Figure 5 polymers-14-03438-f005:**
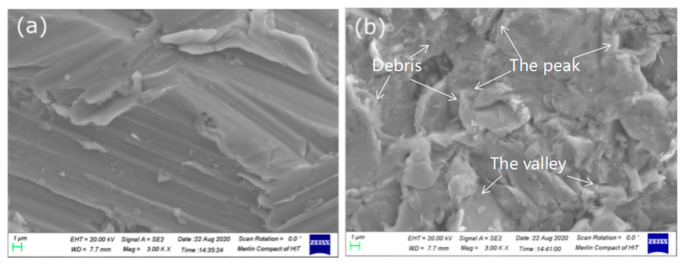
SEM photos of steel surfaces after (**a**) grinding and (**b**) sand blasting.

**Figure 6 polymers-14-03438-f006:**
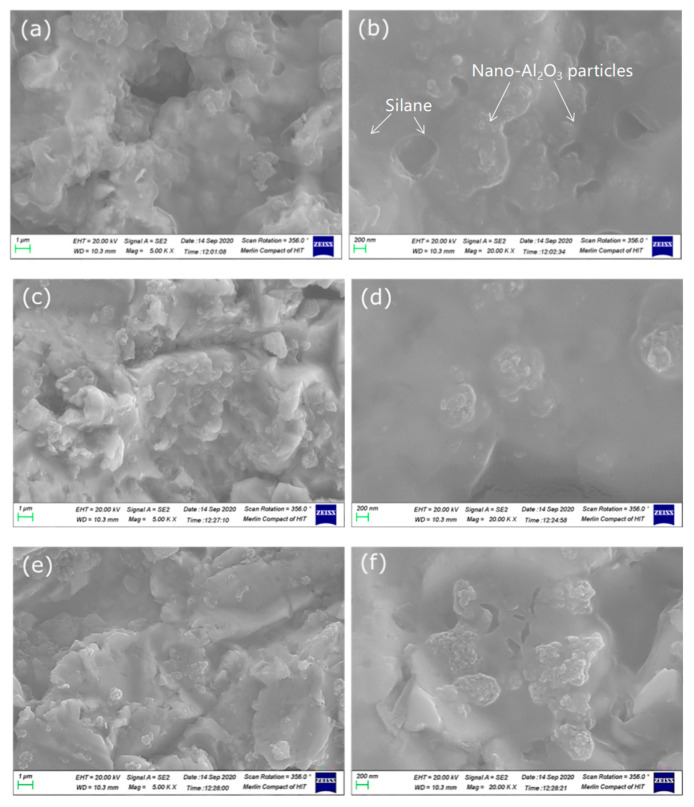
SEM photos of 1 wt.% (**a**,**b**), 2% (**c**,**d**), and 3% (**e**,**f**) silanized Al_2_O_3_ on steel surface.

**Figure 7 polymers-14-03438-f007:**
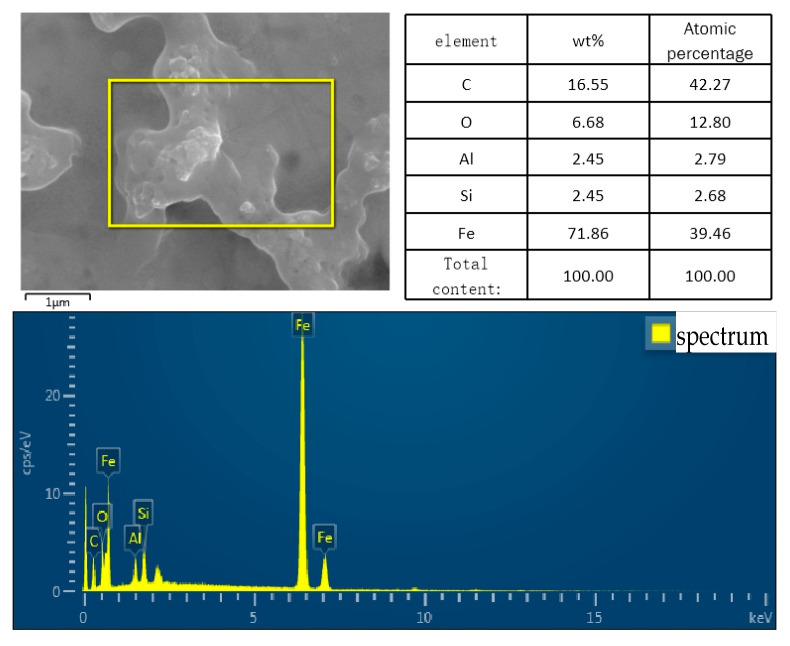
EDS photos of Al_2_O_3_ particles of steel surface.

**Figure 8 polymers-14-03438-f008:**
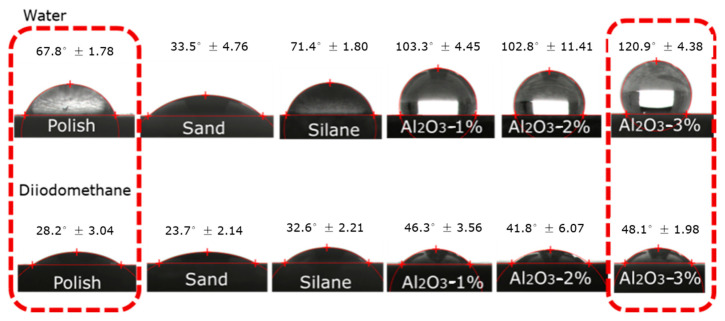
Photos of contact angle tests.

**Figure 9 polymers-14-03438-f009:**
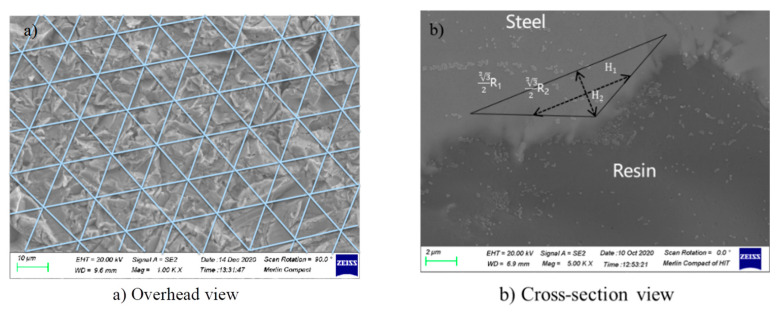
Triangular mesh division of matrix surface after sandblasting (**a**): overhead view and (**b**) cross-cutting perspective.

**Figure 10 polymers-14-03438-f010:**
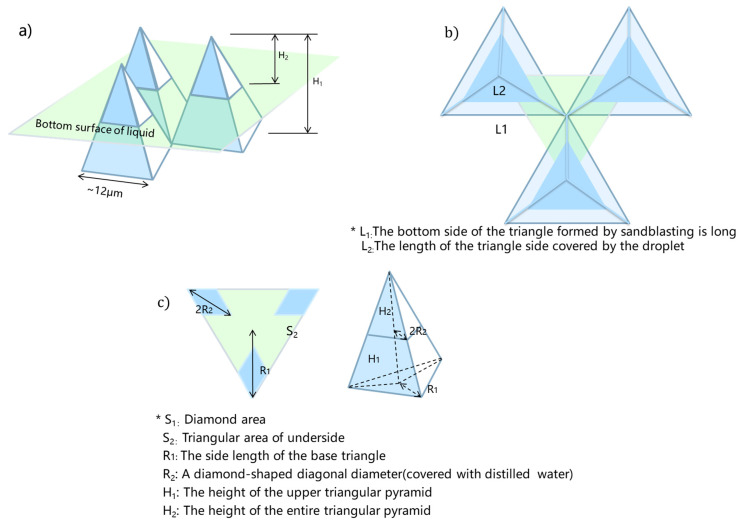
Triangular pyramid model formed by sandblasting: (**a**) Three-dimensional view of the steel surface formed by sandblasting; (**b**) overhead view of steel surface; and (**c**) a triangular pyramid formed by sandblasting (* refers to the parameter of peaks and valley of steel after sandblasting).

**Figure 11 polymers-14-03438-f011:**
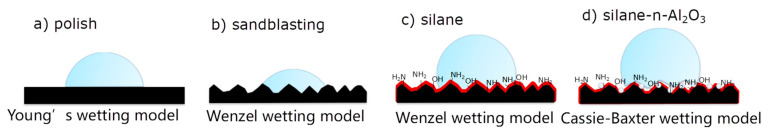
The matrix surface formed by different treatment methods and related models (**a**) polish followed Young’ wetting model (**b**) sandblasting followed Wenzel wetting model (**c**) silane followed Wenzel wetting model (**d**) silane-n-Al_2_O_3_ followed Cassie-Baxter wetting model.

**Figure 12 polymers-14-03438-f012:**
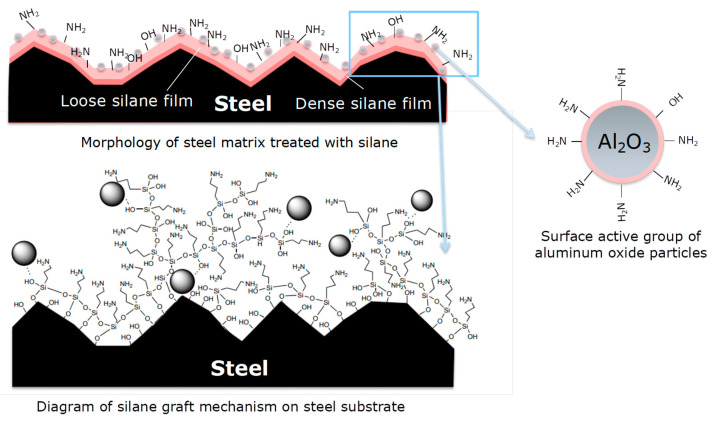
Wettability angle models of steel substrate surface with different treatments.

**Figure 13 polymers-14-03438-f013:**
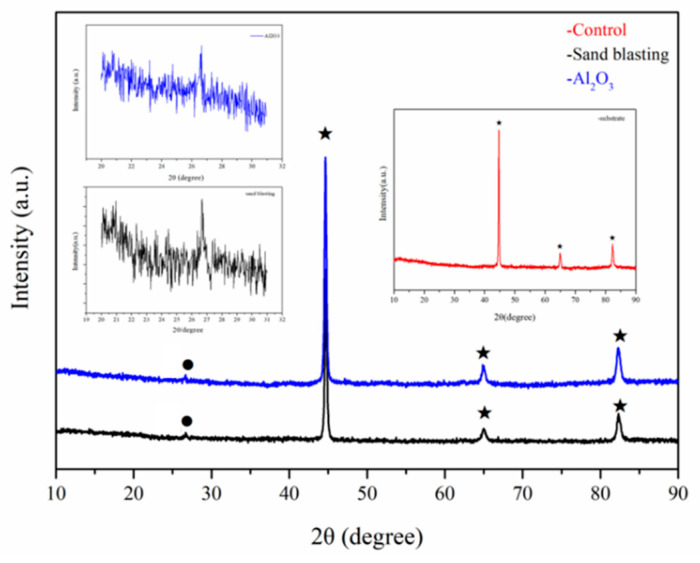
XRD analysis of steel substrate surface. (● refer to Al; ★ refer to Fe).

**Figure 14 polymers-14-03438-f014:**
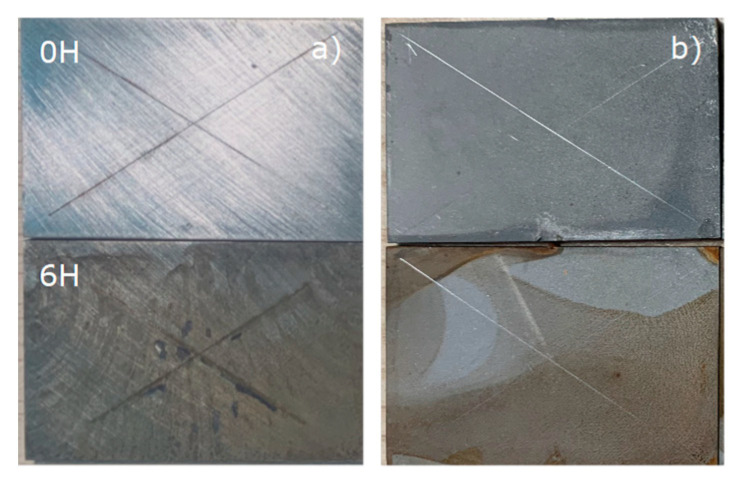
Durability test of steel surface with different treatments: (**a**) control; and (**b**) n-Al_2_O_3_.

**Figure 15 polymers-14-03438-f015:**
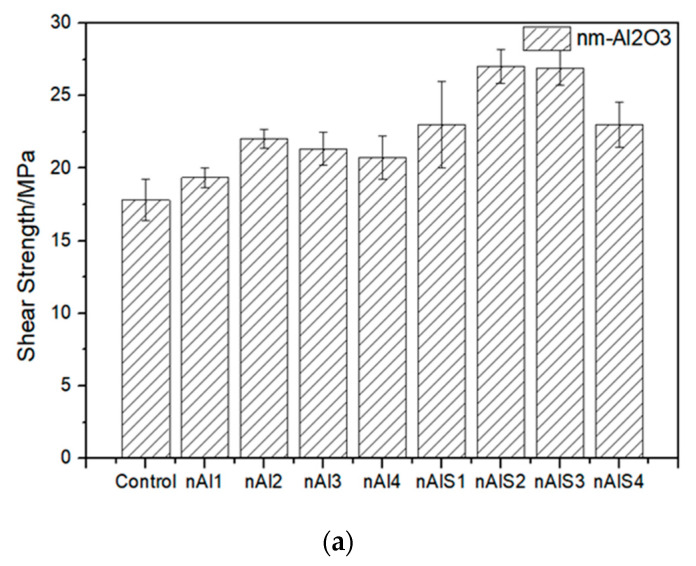
Effect of different contents of (**a**) nano-Al_2_O_3_ particles and (**b**) micro-Al_2_O_3_ particles on the interface properties of steel/resin.

**Figure 16 polymers-14-03438-f016:**
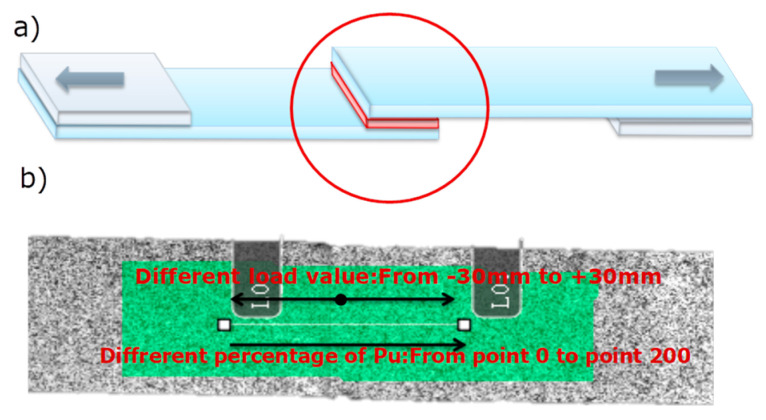
Single-lap sample and speckle on sample surface (**a**) graphic of single-lap joints; (**b**) illustration of single lap joints for DIC.

**Figure 17 polymers-14-03438-f017:**
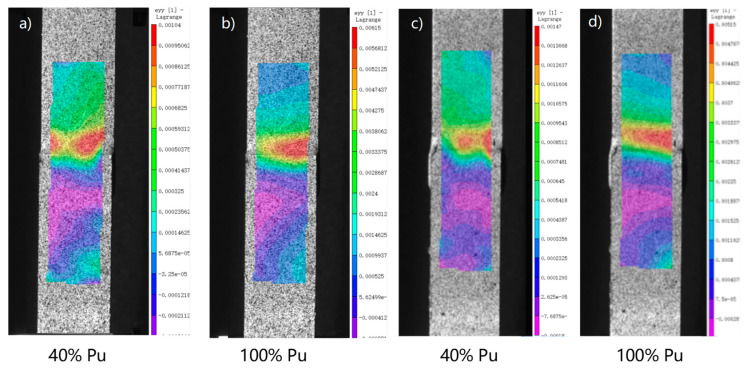
Mechanical properties of single-lap joints with different treatment methods by DIC analysis procedure: Control: (**a**) 40% Pu; (**b**) 100% Pu; 2 wt.% Al_2_O_3_: (**c**) 40% Pu; and (**d**) 100% Pu.

**Figure 18 polymers-14-03438-f018:**
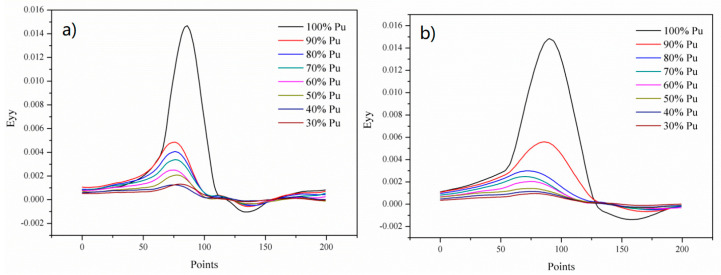
Mechanical properties of single-lap joints with different treatment methods by DIC analysis procedure. (**a**) Control; and (**b**) 2 wt.% Al_2_O_3_.

**Figure 19 polymers-14-03438-f019:**
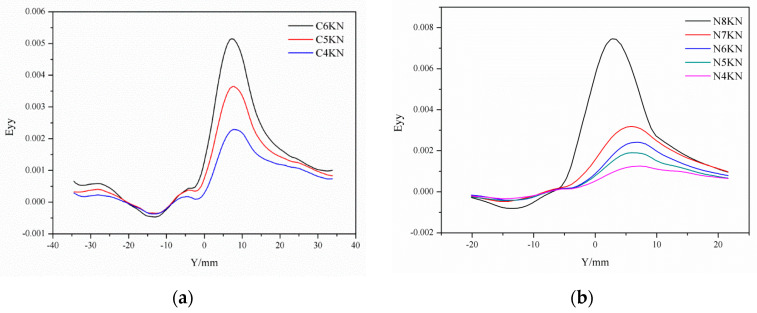
Mechanical properties of single-lap joints with different treatment methods by DIC analysis procedure: (**a**) Control; and (**b**) 2 wt.% Al_2_O_3_.

**Figure 20 polymers-14-03438-f020:**
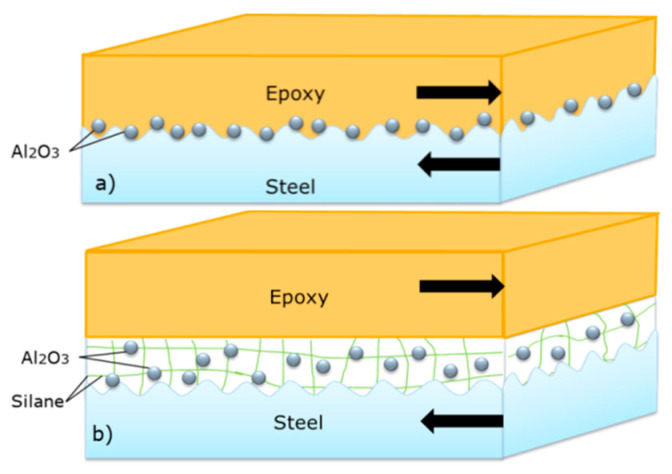
Sketch of epoxy resin and steel surface bonding: (**a**) Nano-Al_2_O_3_ particles on steel surface; and (**b**) silanized Al_2_O_3_ particles on steel surface.

**Figure 21 polymers-14-03438-f021:**
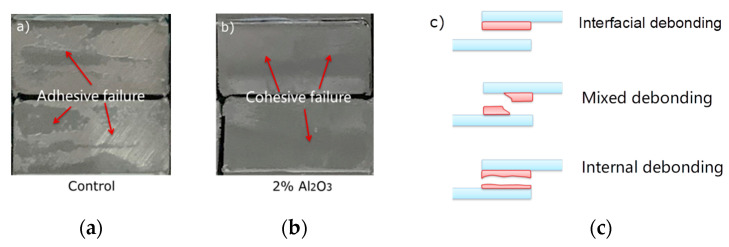
Digital photos of epoxy resin and steel surface: (**a**) Control sample; (**b**) 2% Al_2_O_3_; and (**c**) failure modes.

**Figure 22 polymers-14-03438-f022:**
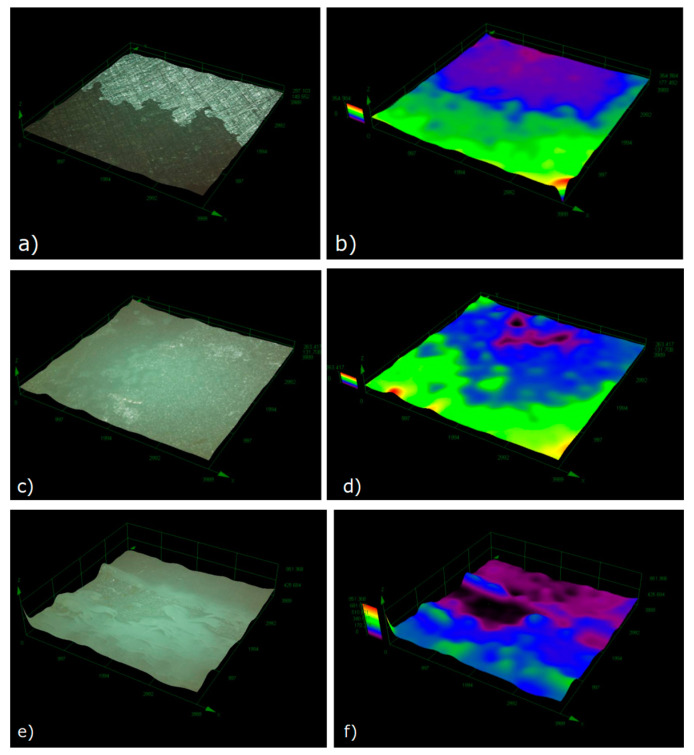
Optical micrographs and 3D analysis of single-lap cross-section fracture: (**a**,**b**) control sample; (**c**,**d**) 1 wt.% nAl_2_O_3_; and(**e**,**f**) 3 wt.% nAl_2_O_3_.

**Table 1 polymers-14-03438-t001:** Preparation procedures of different samples.

	Grind	Sand Blast	Silane	Nano/Micron Al_2_O_3_-Silane	Nano/Micron Al_2_O_3_
Grind	√				
Sand blast	√	√			
Silane	√	√	√		
Al_2_O_3_-silane	√	√		√	
Al_2_O_3_	√	√			√

**Table 2 polymers-14-03438-t002:** FTIR analysis of silane and silanized Al_2_O_3_ solution.

Different Solution		Reference Range (cm^−1^)	Bond Characteristic
Silane	Silanized-CNT		
3300	3300	3570~3050	OH stretching vibration
-	2965	2960~2875	CH stretching vibration
1636	-	1650~1560	-NH_2_
-	1100	1150~1040	C-OH
1048	1040	1100~1000	R-O-Si
	878	955~830	Si-OH

**Table 3 polymers-14-03438-t003:** Effect of different content of nano-Al_2_O_3_ particles.

Sample	Control	nAl1	nAl2	nAl3	nAl4	nAlS1	nAlS2	nAlS3	nAlS4
Shear strength	17.8 ± 1.43	19.35 ± 0.66	22.04 ± 0.65	21.33 ± 1.13	20.74 ± 1.47	23.01 ± 2.96	27.03 ± 1.18	26.9 ± 1.21	22.99 ± 1.55

**Table 4 polymers-14-03438-t004:** Effect of different contents of micro-Al_2_O_3_ particles.

Sample	MAl1	MAl2	MAl3	MAl4	MAlS1	MAlS2	MAlS3	MAlS4	MAl1
Shear strength	21.92 ± 1.2	22.8 ± 0.86	22.16 ± 0.9	22.65 ± 0.45	19.95 ± 0.36	21.56 ± 1.21	20.59 ± 0.41	20.74 ± 0.9	21.92 ± 1.2

## Data Availability

Not applicable.

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
