# Peer review of "A Novel Treatment: Effects of Nano-Sized and Micro-Sized Al2O3 on Steel Surface for the Shear Strength of Epoxy–Steel Single-Lap Joints"

_polymers, 2022, doi:10.3390/polym14173438_

Round 1
Reviewer 1 Report
- Language check needs to be done. I recommend professional proofreading. There are far too many spelling errors including in the title - ‘novol’ instead of novel. Same with grammar.
- Referencing style can be improved. For instance, Page 2, line 67 there is a sentence starting Ismail Sarac et al… but the reference is given in the next sentence. It is better to write as Sarac et al. [24] investigated. Similarly, it should be Akpinar et al. [25]. No need to use full names.
- I am not sure what is the reason behind sandblasting a polished steel. Usually when we want to compare surface treatment A and treatment B, we take one sample and apply treatment A and another sample with treatment B. As supplied material is used as control.
- I may not be fully familiar with the topic but my understanding is that sandblasting and chemical treatments improve the adhesion properties by different methods; mechanical interlocking and changing surface chemistry. I will be more curious to compare sandblasted sample (only mechanical interlocking) to silane treated (only chemical treatment). But the authors have added chemical treatment on top of sandblasting.
- Section 3.1 is very disjointed and is a collection of paragraphs from different articles. There is no narrative cogency.
- The schematic diagrams in fig 4 don’t correspond to the caption. The (a) is not polished and (b) is not of sandblasted, etc. There is no (f).
- The XRD analysis in Fig 13 is not clear to me. If the presence of Al2O3 can be detected by the peak in 27 degrees, shouldn’t the sandblasted sample not have that peak.
- Fig 14 of durability test is too small.
- Fig 15 is missing the micro-Al2O3 results. There is no fig (b)
- The figure numbering in the text is not consistent. The speckle on sample surface figure is called Figure 13 even though it is after Fig.15.
- The DIC analysis does not really add any value to the article. What are you trying to measure or compare here? It looks like you are imaging the SLJ sample face-on instead of side-on at the interface. For instance, look at the setup/ results of Zongkai He, Quantian Luo, Qing Li, Gang Zheng, Guangyong Sun, Fatigue behavior of CFRP/Al adhesive joints — Failure mechanisms study using digital image correlation (DIC) technique, Thin-Walled Structures, Volume 174, 2022
- I do not see what is the point of the results of the DIC analysis in Fig 15 and 16 (should be 18 and 19) as it does not indicate the initiation of failure or show any different failure modes (cohesive vs adhesive failure). It would be more interesting to see a load vs displacement curve for the different treatments.
- There are superscript numbers throughout the text. Either they are references which are not in the right format or they have been copy-pasted from source text and shouldn’t be there.
- The main comment I have is that the paper tries to do too much but not in sufficient depth. Like there are micro particles, then nano particles. Different % of them. With silane, without silane. The story gets diluted. It feels like they just want to show all the work that they did but that is not the purpose of a journal article. I will suggest to have a tighter focus and limit the scope of the article to the grafting of Al2O3 on the surface, contact analysis, micrography, mechanical testing, failure surface. there is half a paragraph about durability. Why? There are 2 lines about XRD analysis. Some figures of DIC. But they are not really informative.
Author Response
- Language check needs to be done. I recommend professional proofreading. There are far too many spelling errors including in the title - ‘novol’ instead of novel. Same with grammar.
A:Accept.Some grammar and spelling mistakes have been corrected.
- Referencing style can be improved. For instance, Page 2, line 67 there is a sentence starting Ismail Sarac et al… but the reference is given in the next sentence. It is better to write as Sarac et al. [24] investigated. Similarly, it should be Akpinar et al. [25]. No need to use full names.
A: Accept. The sentence structure had been modified by using endnote software.
- I am not sure what is the reason behind sandblasting a polished steel. Usually when we want to compare surface treatment A and treatment B, we take one sample and apply treatment A and another sample with treatment B. As supplied material is used as control.
A:The polishing procedure is the pre-treatment for steel to remove the rust on the steel surface. After polishing, the roughness of steel plates needs to be enhanced by sandblasting procedure.
- I may not be fully familiar with the topic but my understanding is that sandblasting and chemical treatments improve the adhesion properties by different methods; mechanical interlocking and changing surface chemistry. I will be more curious to compare sandblasted sample (only mechanical interlocking) to silane treated (only chemical treatment). But the authors have added chemical treatment on top of sandblasting.
A: As the basic treatment of the steel surface, sandblasting could provide the active reaction points on steel surface. And then the steel could be more easily to react with silane. So sandblasting the necessary step for steel surface treatments.
- Section 3.1 is very disjointed and is a collection of paragraphs from different articles. There is no narrative cogency.
The schematic diagrams in fig 4 don’t correspond to the caption. The (a) is not polished and (b) is not of sandblasted, etc. There is no (f).
The XRD analysis in Fig 13 is not clear to me. If the presence of Al2O3 can be detected by the peak in 27 degrees, shouldn’t the sandblasted sample not have that peak.
A: section 3.1(Al2O3 suspension of tests) is the pre-step for application of nano-solution. The label in Fig.4(a-e) has been changed in the paper. (a) sand blasting (b) micro Al2O3 (c) nano Al2O3 (d) micro silanized Al2O3 (e) nano silanized Al2O3. Micro-Al2O3 particles (100 μm) were used to form new peaks and valley on steel surface by sandblasting.
- Fig 14 of durability test is too small.
A: Accept. Several samples were used in the experiment. Only two were showed in the word version.
- Fig 15 is missing the micro-Al2O3 results. There is no fig (b)
A: Accept. Fig.b) has been showed in the new version.
- The figure numbering in the text is not consistent. The speckle on sample surface figure is called Figure 13 even though it is after Fig.15.
A: Accept. The number of figure has been changed in order.
- The DIC analysis does not really add any value to the article. What are you trying to measure or compare here? It looks like you are imaging the SLJ sample face-on instead of side-on at the interface. For instance, look at the setup/ results of Zongkai He, Quantian Luo, Qing Li, Gang Zheng, Guangyong Sun, Fatigue behavior of CFRP/Al adhesive joints — Failure mechanisms study using digital image correlation (DIC) technique, Thin-Walled Structures, Volume 174, 2022
A: Steel is a kind of homogeneous material. According to the 3D technology, the interface deformation condition could been reveal through surface condition of the steel.
- I do not see what is the point of the results of the DIC analysis in Fig 15 and 16 (should be 18 and 19) as it does not indicate the initiation of failure or show any different failure modes (cohesive vs adhesive failure). It would be more interesting to see a load vs displacement curve for the different treatments.
A: Accept. The curve is the analysis curve based on the line on steel surface(Fig.16). It only shows the Eyy(strain deformation) on steel surface instead of the failure mode. The maximum failure load of different treatment is shown in Fig.15. And the failure mode of different treatment could be reveal by Fig.22.
- There are superscript numbers throughout the text. Either they are references which are not in the right format or they have been copy-pasted from source text and shouldn’t be there.
A: Accept. The authors will contact with editors.
- The main comment I have is that the paper tries to do too much but not in sufficient depth. Like there are micro particles, then nano particles. Different % of them. With silane, without silane. The story gets diluted. It feels like they just want to show all the work that they did but that is not the purpose of a journal article. I will suggest to have a tighter focus and limit the scope of the article to the grafting of Al2O3 on the surface, contact analysis, micrography, mechanical testing, failure surface. there is half a paragraph about durability. Why? There are 2 lines about XRD analysis. Some figures of DIC. But they are not really informative.
A: The paper reveals the grafting procedure, the possible effects of treatments and the results of different treatments. The durability tests could be regarded as part of results of different treatments, due to surface treatment of steel easily be destroyed by the moisture in environment. Other information has been well organized.
Reviewer 2 Report
The current paper pertains to an interesting topic. Namely, a study regarding the effects of nano-sized and micro-sized Al2O3 on steel surface for the shear strength of epoxy-steel single lap joints is presented. In general, it is well written, nevertheless, there are some issues that have to be pointed out and improved.
More specifically:
· in page 1 and line 33 the phrase "...suffer from some advantages of..." does not make sense. Please correct it accordingly.
· an overall grammar and syntax improvement is suggested.
· Figures must be referred in the text by their numbering order. For example, you cannot refer Figure 8 (see page 4 and line 137) before Figure 3.
· No information regarding the shear test measuring method is included. Please provide some more details (e.g., number of tests, followed ISO, etc.).
· In general, the term "polishing" is used for machining surfaces of significant low roughness. After polishing the sandblasting could resulted a degradation of the surface quality and not, as authors state in line 216, to highly improve the surface. Considering this, I suggest to replace the term polishing by the more general term grinding.
· although a significant detailed analysis of the contact angle and the wettability of the surface is presented it is not clearly explained how the wettability is related to the shear strength. Thus, please provide some more apt connection between the contact angle results and the shear strength of the joint.
· there are two Figures 14.
· Figures quality needs improvement.
· please provide the data of Figure 15 in the form of a Table.
· The References are not in line with journal template
Considering the aforementioned, the current paper can be accepted only after a major revision.
Author Response
The current paper pertains to an interesting topic. Namely, a study regarding the effects of nano-sized and micro-sized Al2O3 on steel surface for the shear strength of epoxy-steel single lap joints is presented. In general, it is well written, nevertheless, there are some issues that have to be pointed out and improved.
More specifically:
- in page 1 and line 33 the phrase "...suffer from some advantages of..." does not make sense. Please correct it accordingly.
A: Accept. It has been corrected.
- an overall grammar and syntax improvement is suggested.
A: Accept. The mistakes has been corrected.
- Figures must be referred in the text by their numbering order. For example, you cannot refer Figure 8 (see page 4 and line 137) before Figure 3.
A:Accept. The mistake has been corrected.
- No information regarding the shear test measuring method is included. Please provide some more details (e.g., number of tests, followed ISO, etc.).
A: Accept. The shear strength test followed ASTM D 1002.
- In general, the term "polishing" is used for machining surfaces of significant low roughness. After polishing the sandblasting could resulted a degradation of the surface quality and not, as authors state in line 216, to highly improve the surface. Considering this, I suggest to replace the term polishing by the more general term grinding.
A: Accept. It has been corrected.
- although a significant detailed analysis of the contact angle and the wettability of the surface is presented it is not clearly explained how the wettability is related to the shear strength. Thus, please provide some more apt connection between the contact angle results and the shear strength of the joint.
A: Accept. The contact angle related to wettability between steel and adhesive. And the wettability is one of the key factors of shear strength.
- there are two Figures 14.
A: Accept. The number of Figures have been changed.
- Figures quality needs improvement.
A: Accept. Some of them with poor quality.
- please provide the data of Figure 15 in the form of a Table.
A: The table were listed in the paper.
- The References are not in line with journal template
A: It has been changed.
Considering the aforementioned, the current paper can be accepted only after a major revision.
Reviewer 3 Report
In my opinion, the article looks interesting, a number of studies have been carried out, but when reading the paper, there are several elements missing:
1. One of the missing parts is the discussion of the results. Without it, the paper looks more like an extended research report, not a scientific paper. In the discussion I would expect a reference to works on similar subjects, to results obtained by other authors.
2. Secondly, perhaps this information is included in the text and I missed it while reading, I mean the number of repeats for each part of surface preparation. How many samples were tested? This information should be included in section 2.2 Steel surface treatment or 2.3 Preparation of single lap joints.
3. Lack of clarity in the text. In the paper, in section 3, the authors describe the method of testing (section 2.4 - 2.6). The same order of testing should be preserved in the description of obtained results in point 3. Moreover, there is a lack of information concerning details of strength testing. There is only a mention in line 349: "The micro-particles were also tested by universal tensile testing machine". For me, this information is insufficient.
4. Another thing I find missing in the article is a statistical analysis of the strength results obtained. In Fig. 15 it should be indicated whether the results presented concern individual samples or average values. What do the error bars in the diagram mean? The bars overlap, so we cannot clearly determine if there are differences between the individual values. The authors should supplement the paper with a statistical analysis that determines the significance of the differences for the individual groups of samples tested.
5. There are some single editing and interpunctuation errors, e.g. missing spaces or extra brackets, e.g. in lines: 60, 62, 64, 65, 70, 77, 258, etc. Errors of this type should be checked and corrected throughout the paper.
6. What does superscript mean e.g. in line 46 at properties, line 53 at % or in lines 400, 411, 419?
Author Response
In my opinion, the article looks interesting, a number of studies have been carried out, but when reading the paper, there are several elements missing:
- One of the missing parts is the discussion of the results. Without it, the paper looks more like an extended research report, not a scientific paper. In the discussion I would expect a reference to works on similar subjects, to results obtained by other authors.
A: Accept. The comparisons with other authors could be added into the discussion part.
- Secondly, perhaps this information is included in the text and I missed it while reading, I mean the number of repeats for each part of surface preparation. How many samples were tested? This information should be included in section 2.2 Steel surface treatment or 2.3 Preparation of single lap joints.
A:Accept. 5 samples were tested in each condition.
- Lack of clarity in the text. In the paper, in section 3, the authors describe the method of testing (section 2.4 - 2.6). The same order of testing should be preserved in the description of obtained results in point 3. Moreover, there is a lack of information concerning details of strength testing. There is only a mention in line 349: "The micro-particles were also tested by universal tensile testing machine". For me, this information is insufficient.
A: Accept. More information has been added into the new version.
- Another thing I find missing in the article is a statistical analysis of the strength results obtained. In Fig. 15 it should be indicated whether the results presented concern individual samples or average values. What do the error bars in the diagram mean? The bars overlap, so we cannot clearly determine if there are differences between the individual values. The authors should supplement the paper with a statistical analysis that determines the significance of the differences for the individual groups of samples tested.
A: Accept. A Table with the details of test results has been given to the new version.
- There are some single editing and interpunctuation errors, e.g. missing spaces or extra brackets, e.g. in lines: 60, 62, 64, 65, 70, 77, 258, etc. Errors of this type should be checked and corrected throughout the paper.
A: Accept.The grammar mistakes were corrected.
- What does superscript mean e.g. in line 46 at properties, line 53 at % or in lines 400, 411, 419?
A: The format mistakes have been modified.
Round 2
Reviewer 1 Report
The manuscript has been improved to reflect the comments of the reviewer. It can be accepted for publication
Reviewer 2 Report
Authors have significantly improve the manuscript according to the review comments, thus, in my humble opinion, it can be accepted in present form.
Reviewer 3 Report
Paper accept in present form.